# Nitrogen Fertilization Shapes Soil Microbial Diversity and Ecosystem Multifunctionality by Modulating Soil Nutrients

**DOI:** 10.3390/microorganisms13030540

**Published:** 2025-02-27

**Authors:** Yaoquan Zhang, Zhuzhu Luo, Liangliang Li, Lili Nian, Lingling Li, Yining Niu, Renyuan He, Jiahe Liu

**Affiliations:** 1College of Forestry, Gansu Agricultural University, Lanzhou 730070, China; zhangyqgs@163.com (Y.Z.); 18393414923@163.com (R.H.); 2College of Resources and Environmental Sciences, Gansu Agricultural University, Lanzhou 730070, China; liujiahe@gsau.edu.cn; 3Grassland Science College, Gansu Agricultural University, Lanzhou 730070, China; 18394797671@163.com; 4Institute of Soil, Fertilizer and Water-Saving Agriculture, Gansu Academy of Agricultural Sciences, Lanzhou 730070, China; nll18893814845@163.com; 5State Key Laboratory of Aridland Crop Science, Gansu Agricultural University, Lanzhou 730070, China; lill@gsau.edu.cn (L.L.); niuyn@gsau.edu.cn (Y.N.)

**Keywords:** alfalfa, ecosystem multifunctionality, loess plateau, nitrogen fertilizer addition, soil microorganisms

## Abstract

Soil microbial communities play an important role in driving diverse ecosystem functions and ecological processes and are the main driving force for maintaining biogeochemical cycles. To investigate the effects of nitrogen fertilizer addition on soil microbial community characteristics and ecosystem multifunctionality in alfalfa fields, a field experiment was conducted in the semi-arid region of the Loess Plateau. Ecological network analysis revealed a strong cooperative relationship among bacterial community species under the N100 treatment, while a strong competitive relationship was observed among fungal community species under the N50 treatment. Furthermore, compared with the control check, the soil carbon nutrient function, ecosystem multifunctionality and grassland productivity of N150 treatment increased by 45.17%, 34.01%, and 7.92%, while the soil phosphorus function decreased by 13.44%. Additionally, soil pH significantly influences ecosystem multifunctionality, soil carbon nutrient function, and grassland productivity. Soil water content notably affects the soil phosphorus nutrient function, while soil microbial diversity has a significant impact on grassland productivity and soil potassium nutrient function. The above results suggest that alterations in soil nutrient levels influence ecosystem multifunctionality by regulating microbial community diversity, offering new insights into the mechanisms by which nutrients impact soil microbial communities and ecosystem properties.

## 1. Introduction

The Loess Plateau is one of the most ecologically fragile regions in China. Excessive land use has been a major contributor to the destruction of natural vegetation, soil erosion, biodiversity loss, and the decline of ecosystem functions in this area. In order to improve this situation, vegetation restoration and soil improvement have become the focus of attention of the local government and various sectors of society [1]. Alfalfa (*Medicago sativa* L.), a legume with good ecological adaptability and stress resistance, is widely used in vegetation restoration and soil improvement projects on the Loess Plateau [2]. Studies have shown that planting alfalfa can improve soil physical and chemical properties, change the structure of soil microbial communities, and thus affect ecosystem functions [3]. Soil microorganisms, including archaea, bacteria, and fungi, are microscopic organisms invisible to the naked eye [4]. These microorganisms play a pivotal role in soil ecosystems, serving as key indicators of ecosystem health, soil quality, and the progress of land restoration. Moreover, they are integral to ecosystem functions, driving processes such as plant productivity, soil nutrient cycling, and energy flow [5,6]. Ecosystem multifunctionality (EMF) refers to the ability of an ecosystem to simultaneously provide multiple functions and services [7]. As a comprehensive indicator for evaluating multiple ecological functions, it is of great significance for a comprehensive understanding of the structure and function of the ecosystem. In recent years, EMF has emerged as a key focus in ecological research. However, there is limited research on the mechanisms by which soil microorganisms influence the EMF of alfalfa under nitrogen fertilizer addition.

Nitrogen addition not only alters the physical and chemical properties of soil but also impacts the functions of soil microbial communities. This can inhibit the growth of certain microorganisms while promoting the rapid proliferation of microorganisms with specific ecological functions [8]. Following nitrogen fertilizer application, the smaller the fluctuation range of the number of microorganisms, population structure, and various biogenic elements in the soil in the short term, the stronger the resilience and resistance of the soil ecosystem, the better its health [9]. Additionally, nitrogen fertilizer application enhances plant productivity and promotes crop growth [10]. Soil microorganisms play a crucial role in promoting nutrient cycling and transformation through processes such as nitrogen fixation, nitrification, and denitrification under nitrogen fertilizer conditions, thereby providing plants with sufficient nitrogen sources [11]. However, excessive nitrogen fertilizer application can result in nutrient surpluses, leading to issues such as water eutrophication and soil acidification, which severely compromise the stability and sustainability of ecosystem functions [12]. In recent years, advances in techniques such as phospholipid fatty acid (PLFA) analysis, BIOLOG physiological profiling, and especially molecular biology, have provided a comprehensive understanding of microbial community structure changes under various conditions. Microbial community diversity serves as a critical indicator, indirectly reflecting community structure and function. Consequently, these biotechnologies are widely applied to study the effects of fertilization practices on soil microbial community structure and functional diversity. In summary, nitrogen fertilizer addition has complex impacts on soil microbial diversity and ecosystem function. The reasonable application of nitrogen fertilizer, along with efforts to maintain soil microbial diversity and ecosystem function stability, is essential for achieving both agricultural productivity and ecological sustainability.

In natural ecosystems, both controlled indoor experiments [13] and in situ surveys [14] have demonstrated a positive correlation between soil microbial diversity and ecosystem multifunctionality (EMF). Thus, as soil microbial diversity recovers in degraded ecosystems, ecosystem functions also improve. However, compared to studies on ecosystems such as farmland [15], research on the relationship between soil microbial diversity and EMF in restoration efforts involving alfalfa cultivation remains limited. Furthermore, as research progresses, it has become evident that soil microorganisms do not exist in isolation but interact through processes such as nutrient cycling, energy flow, and information exchange, forming complex ecological networks. These networks are critical for maintaining ecosystem functions [16]. Therefore, focusing solely on microbial diversity is insufficient to fully understand their mechanisms of action within ecosystems. As a standardized research framework, co-occurrence network analysis offers a robust method for studying microbial community structures and their interactions. This approach enables a deeper understanding of the relationships among microbial groups, as well as their functions and roles within ecosystems. Consequently, co-occurrence network analysis has become widely utilized in microbial ecology research, providing a powerful tool to reveal the complex interactions within microbial communities [17].

Since the implementation of China’s Grain for Green Program, the ecological environment of the Loess Plateau has significantly improved, leading to increased vegetation coverage, enhanced soil physicochemical properties, enriched soil microbial communities, and restored ecosystem functions. This study utilizes a long-term (8-year) nitrogen fertilizer gradient experiment conducted in alfalfa fields on the Loess Plateau. High-throughput sequencing techniques were employed to systematically assess the effects of nitrogen addition on alfalfa growth and rhizosphere microbial communities. The study tested three hypotheses: (1) Nitrogen fertilizer addition enhances the cooperative relationship between soil bacterial species while reducing that between fungal species; (2) Nitrogen fertilizer addition leads to a decrease in soil pH, indirectly affecting microbial diversity and, consequently, ecosystem functions; (3) The nitrogen nutrition ecosystem function (EF-N) is the primary driver of soil microbial diversity. Furthermore, this study examines soil microbial diversity (bacteria and fungi), the complexity of microbial co-occurrence networks, and ecosystem multifunctionality, while exploring how soil nutrients mediate the relationship between microbial diversity and ecosystem functions. The findings aim to provide theoretical insights for improving soil health and restoring ecosystem functions in the Loess Plateau.

## 2. Material and Methods

### 2.1. Site Description

The experiment was conducted at the Comprehensive Experiment Station of Dryland Agriculture, Gansu Agricultural University, located in Lijiabao Town, Anding District, Dingxi City, Gansu Province (35°28′ N, 104°44′ E) (Figure 1). The experimental site is situated in the mid-temperate semi-arid zone, characterized by an average annual sunshine duration of 2476.6 h, average annual solar radiation of 592.9 kJ·cm^−2^, an accumulated temperature of ≥0 °C totaling 2933.5 °C, and an accumulated temperature of ≥10 °C of 2239.1 °C. The region experiences a frost-free period of 140 days, an average annual temperature of 6.4 °C, average annual precipitation of 390 mm, and average annual evaporation of 1531 mm. This area exemplifies a typical one-crop-per-year dryland, rain-fed agricultural zone.

### 2.2. Experimental Design and Soil Sampling

The experiment began on 6 April 2014, with alfalfa established by strip sowing at a seeding rate of 22.5 kg/ha^2^. Ten rows were sown in each plot, with a row spacing of 30 cm. Four nitrogen fertilizer treatments were applied: 0 kg/ha^2^ (CK), 50 kg/ha^2^ (N50), 100 kg/ha^2^ (N100), and 150 kg/ha^2^ (N150). Each treatment was repeated 3 times, with a total of 12 plots, randomly arranged, and the area of each plot was 12 m^2^ (3 m × 4 m). Nitrogen fertilizer was applied annually, using urea (N46%) as the test fertilizer. The experiment was conducted under fully rain-fed conditions, with no irrigation.

Soil samples were collected in June 2021 during the bloom period of the first alfalfa crop. Five alfalfa plants were randomly selected from each plot, and soil samples were collected from the 0–20 cm depth around the alfalfa roots using a soil drill. The soil was mixed into a single composite sample, and debris such as stones, gravel, and plant residues were removed. Each soil sample was then divided into two parts: one portion was stored in a freezer at 4 °C for the determination of microbial biomass carbon, nitrogen, and phosphorus, while the other was air-dried for the analysis of soil physicochemical properties. The remaining portion was used for the characterization of soil microbial communities, with microbial measurements conducted only in 2021.

### 2.3. Analysis of Soil Physicochemical Properties

In this study, selected soil properties were measured, including soil moisture, pH, soil organic carbon, total nitrogen, total phosphorus, available phosphorus, available potassium, nitrate nitrogen, ammonium nitrogen, microbial biomass carbon, microbial biomass nitrogen, and microbial biomass phosphorus. Soil moisture was determined by the drying method (105 °C for 24 h) [18]. Soil pH was measured using a pH meter (Mettler Toledo, Zurich, Switzerland) at a soil-to-water ratio of 1:2.5. soil organic carbon was analyzed using the dichromate oxidation method [19]. Total nitrogen was determined using the Kjeldahl method [20]. Total phosphorus was determined by the molybdenum blue method [21]. Available phosphorus was extracted with 0.5 M NaHCO_3_ (pH = 8.5) and determined using the molybdenum blue method [21]. Analysis of nitrate nitrogen and ammonium nitrogen was conducted using a continuous flow analyzer [22]. Available potassium content was determined by NH_4_OAc leaching and flame photometry [23]. microbial biomass carbon and microbial biomass nitrogen were determined using an ACN 802 carbon and nitrogen analyzer (VELP Scientific, Usmate, Italy), while microbial biomass phosphorus was extracted with 0.5 M NaHCO_3_ and determined using the molybdenum blue method [24].

### 2.4. DNA Extraction and High-Throughput Sequencing

Total soil DNA was extracted from 0.25 g of fresh soil using the PowerSoil^®^ DNA Isolation Kit (MOBIO Laboratories, Carlsbad, CA, USA) following a standardized method. The DNA extracts were separated and purified on a 1% agarose gel, and the concentration and purity of the extracted DNA were measured using a NanoDrop UV-Vis spectrophotometer (ND-2000c, NanoDrop Technologies, Wilmington, DE, USA). Primers 515F (5′-GTGCCAGGCGCCGCGCGGTA-3′) and 907R (5′-CCGTCAATTCCTTGAGTTT-3′) were used to amplify bacterial 16S rRNA, while primers ITS5-1737F (5′-GGAAGTAAAAGTCGTAACAAGG-3′) and ITS2-2043R (5′-GCTGCGTTCTTCATCGATGC-3′) were used to amplify fungal ITS regions [25]. The amplified products were sent to the Majorbio Bio-Pharm Technology Co., Ltd. (Shanghai, China) for high-throughput sequencing using the Illumina MiSeq sequencing platform.

The raw sequences were optimized using the DADA2 software package [26], and low-quality sequences were trimmed with the Cutadapt software 2.0.dev1 [27]. Sequencing errors were identified using the learnErrors function in DADA2. Overlapping regions of the sequences were merged using the mergePairs function to reconstruct the original data. Chimeric sequences were detected and removed, resulting in high-resolution amplicon sequence variants (ASVs). All ASVs were classified and annotated using the SILVA 138 and UNITE8 databases to determine the taxonomic information for each representative ASV sequence. The soil microbial multi-diversity index was calculated using the mean value method [28]. The abundance of bacteria and fungi was standardized to ensure that the abundance of soil microbial groups fell within the range of 0 to 1. The mean of the standardized bacterial and fungal abundance values was then calculated to derive the soil microbial multi-diversity index.

### 2.5. Determination of Ecosystem Multifunctionality

This study measured 15 ecosystem function (EF) indicators and categorized them into six ecosystem function groups: (1) Soil carbon nutrients and storage (EF-C): soil organic carbon and soil microbial biomass carbon. (2) Soil nitrogen nutrients and storage (EF-N): soil total nitrogen, soil ammonium nitrogen, soil nitrate nitrogen, and soil microbial biomass nitrogen. (3) Soil phosphorus nutrients and storage (EF-P): soil total phosphorus, soil available phosphorus, and soil microbial biomass phosphorus. (4) Soil potassium nutrients and storage (EF-K): soil available potassium. (5) alfalfa primary productivity (EF-GP): alfalfa yield. Soil ecosystem multifunctionality (EMF), including all 11 ecosystem function indicators. In addition, including 2 soil physicochemical properties (soil water content and pH) and 2 soil microbial diversity indices (bacteria and fungi). These indicators were selected for their ability to regulate and maintain key ecological processes in alfalfa ecosystems and because they are widely used in research on ecosystem function and multifunctionality [14,29,30]. The ecosystem multifunctionality index was calculated using the average method [7,31]. First, the 11 ecosystem function indicators were standardized using the formula:(1)$$f_{ij}\;=\frac{x_{ij}-{min}_{ij}}{\;{max}_{ij}-{min}_{ij}}$$
where$\;f_{ij}$ is the standardized value of the j-th ecosystem function variable for plot i, $x_{ij}$ is the actual measured value of the j-th ecosystem function variable for plot i, ${min}_{ij}$ is the minimum value of the j-th ecosystem function variable for the same factor, and ${max}_{ij}$ is the maximum value of the j-th ecosystem function variable among all plot for the same factor.

The individual function method was used to calculate the ecosystem function index (EF) as follows:(2)$${EF}_{ij}\;=\sum_{j}^{n}\frac{f_{ij}}{n}\;$$

The average method is used to calculate the ecosystem multifunctionality index EMF:(3)$${EMF}_{i}\;=\;\frac{1}{N}\sum_{1}^{N}f_{ij}$$
where ${EF}_{ij}$ is the functional index of the j-th function of plot i. n is the number of ecosystem variable indicators included in this function. ${EMF}_{i}$ is the ecosystem multifunctionality index of plot i, calculated as the standardized average of all variable indicators for the plot. N is the total number of ecosystem functions in plot i.

### 2.6. Statistical Analysis

SPSS 26.0 and Excel 2010 were used to process data on soil physicochemical properties and bacterial and fungal community composition. The significance of differences was analyzed using one-way analysis of variance (ANOVA) and multiple comparisons (LSD method, *p* = 0.05). Bacterial and fungal community composition and diversity were analyzed using the Sanger Cloud Platform provided by Meiji Company (Shanghai, China). Molecular ecological networks were constructed using R language R1.6.0 and visualized with Gephi 0.10.1 software. Redundancy analysis (RDA) was performed to assess the relationships between bacterial, fungal, and soil environmental factors, with CANOCO 5.0 used for plotting and Adobe Illustrator CS6 for chart modification.

## 3. Results

### 3.1. Environmental Factors

Significant differences were observed in soil physicochemical indicators under varying nitrogen fertilizer additions (Figure 2). Soil pH, organic carbon, available phosphorus, and available potassium were significantly higher in the CK treatment compared to other treatments (*p* < 0.05) and decreased with increasing nitrogen fertilizer addition. Conversely, soil total nitrogen, nitrate nitrogen, microbial biomass carbon, microbial biomass nitrogen, and microbial biomass phosphorus were significantly lower in the CK treatment than in other treatments (*p* < 0.05) and increased with higher nitrogen fertilizer addition. However, no significant differences were found in soil moisture content, total phosphorus, or ammonium nitrogen among the treatments.

### 3.2. Soil Microbial Community Composition and Diversity

High-throughput sequencing revealed that *Actinobacteriota*, *Acidobacteriota*, *Proteobacteria*, and *Chloroflexi* were the dominant bacterial phyla (Figure 3). The total relative abundance of these phyla under different nitrogen fertilizer addition levels was 74.55% (CK), 74.30% (N50), 74.58% (N100), and 70.93% (N150), respectively, indicating a declining trend with increasing nitrogen fertilizer addition. Among these, the relative abundance of *Actinobacteria* was lowest in the N150 treatment (24.12%) and highest in the N100 treatment (27.25%). In contrast, *Proteobacteria* and *Acidobacteriota* exhibited their highest relative abundances in the CK treatment (19.61% and 19.70%, respectively). *Ascomycota* and *Mortierellomycota* were the dominant fungal phyla, with their combined relative abundances at different nitrogen fertilizer addition levels being 92.14% (CK), 88.93% (N50), 91.02% (N100), and 87.93% (N150). This indicates that the relative abundance of dominant fungal phyla decreased with increasing nitrogen fertilizer addition. Specifically, the relative abundance of *Ascomycota* was lowest in the N150 treatment (62.04%) and highest in the CK treatment (70.88%). Conversely, the relative abundance of *Mortierellomycota* was highest in the N150 treatment (25.88%) and lowest in the CK treatment (21.26%).

Further soil microbial LefSe analysis was conducted to identify microbial communities with significant differences across the four nitrogen fertilizer addition treatments. At the phylum level, bacterial taxa such as *Actinobacteriota* and *Chloroflexi*, as well as fungal taxa like *Glomeromycota*, showed significant differences among treatments (*p* < 0.05). At the genus level, bacterial genera such as *Chitinophaga*, *Arenimonas*, *Microvirga*, *Nitrosospira*, and Streptomyces, as well as fungal genera like *Claroideoglomus*, *Phaeomycocentrospora*, and *Septoglomus*, exhibited significant differences across treatments (*p* < 0.05).

Principal coordinate analysis (PCA) based on the Bray–Curtis algorithm was applied to evaluate structural changes in soil bacterial and fungal communities and to assess the effects of different nitrogen fertilizer additions on their structures (Figure 4). PCA divided the soil bacterial community into two groups: N50 and (CK, N100, and N150). The bacterial community structure of N50 was distinct from that of the other treatment groups, indicating structural differences in soil bacterial communities under varying nitrogen fertilizer levels. Similarly, PCA divided the soil fungal community into two groups: (N50 and N150) and (CK and N100). The fungal community structure of N50 and N150 was distinct from that of CK and N100, highlighting differences in fungal community structure under different nitrogen fertilizer additions. Further analysis using the ANOSIM test revealed that nitrogen fertilizer addition significantly affected the bacterial community (*p* < 0.05) but had no significant effect on the fungal community.

The diversity of soil bacteria showed a decreasing trend with the increase in nitrogen fertilizer addition, with the bacterial diversity of the N50 treatment being higher than that of the other treatments, while the bacterial diversity of the N150 treatment was lower. In contrast, fungal diversity increased with the addition of nitrogen fertilizer. The fungal diversity of the N50 treatment was lower than that of the other treatments, and the fungal diversity of the N150 treatment was higher. Additionally, the comprehensive diversity index of soil microorganisms decreased with increasing nitrogen fertilizer addition, with the CK treatment exhibiting the highest diversity index compared to the other treatments. The richness of soil bacteria and fungi increased with nitrogen fertilizer addition, with the bacterial and fungal richness in the N50 treatment being lower than that of the other treatments, and the richness in the N150 treatment being higher. Moreover, the comprehensive richness index of soil microorganisms increased with nitrogen fertilizer addition, and the N150 treatment exhibited the highest microbial richness (Appendix A).

The characteristics of the soil microbial co-occurrence network varied significantly under different nitrogen fertilizer additions (Figure 5). Analysis of the soil bacterial network revealed that, in the N100 treatment, the number of edges (6690), nodes (500), average weighted degree, and graph density were the highest. This indicates that the complexity of the soil bacterial co-occurrence network in this treatment was greater, with more complex interactions between species. Additionally, the proportion of positively correlated edges in the N100 treatment was higher than in the other treatments, while the proportion of negatively correlated edges was lower, suggesting a stronger collaborative relationship among species in the bacterial community under this treatment. Analysis of the soil fungal network revealed that, in the CK, the number of edges (5812), nodes (200), average weighted degree, and graph density were the highest. This indicates that the complexity of the soil fungal co-occurrence network in the CK was relatively high, with more intricate interactions between species. Additionally, the proportion of positively correlated edges in the CK treatment was higher than in the other treatments, while the proportion of negatively correlated edges was lower. In contrast, the N50 treatment showed a lower proportion of positively correlated edges, and a higher proportion of negatively correlated edges compared to the other treatments, suggesting that nitrogen fertilizer addition promoted a stronger competitive relationship among species in the fungal community.

### 3.3. Multifunctionality of Soil Ecosystems

ANOVA was conducted to evaluate the effects of different nitrogen fertilizer additions on the individual functions of the alfalfa ecosystem. The results are shown in Figure 6. Nitrogen fertilizer addition at the N150 level resulted in higher soil carbon nutrient (EF-C), soil nitrogen nutrient (EF-N), and ecosystem multifunctionality (EMF) compared to other treatments. Conversely, the N150 treatment showed lower values for soil potassium nutrient (EF-K) and grassland productivity (EF-GP) than the other treatments, while soil phosphorus nutrient (EF-P) was significantly lower in the N100 treatment. No significant differences were observed in EF-C, EF-P, and EF-GP among the treatments, whereas significant differences were found for EF-N, EF-K, and EMF (*p* < 0.05).

To better understand the pathways through which nitrogen fertilizer addition impacts ecosystem multifunctionality (Figure 7f), structural equation modeling (SEM) was conducted. The analysis revealed that nitrogen fertilizer directly affects grassland ecosystem multifunctionality, showing a negative but non-significant effect (*p* > 0.05). Additionally, nitrogen fertilizer influences ecosystem multifunctionality indirectly through soil pH. In the SEM of EF-C (Figure 7a), nitrogen fertilizer had a direct, highly significant negative effect on EF-C (*p* < 0.05). Furthermore, it indirectly exerted a highly significant negative effect on EF-C by altering soil pH (*p* < 0.05). In the SEM of soil EF-N (Figure 7b), nitrogen fertilizer directly affected EF-C with a highly significant positive effect (*p* < 0.05). However, the indirect effect of nitrogen fertilizer addition on EF-C was not significant (*p* > 0.05). For the SEM of EF-P (Figure 7c), nitrogen fertilizer directly caused a non-significant negative effect on EF-P (*p* > 0.05), whereas soil water content exerted a very significant negative effect on EF-P (*p* < 0.05). In the SEM of EF-K (Figure 7d), nitrogen fertilizer directly had a significant negative effect on EF-K (*p* < 0.05). Moreover, it indirectly affected EF-K through a reduction in soil pH, which in turn impacted soil microbial diversity, resulting in a significant positive effect on EF-K (*p* < 0.05). Regarding the SEM of EF-GP (Figure 7e), nitrogen fertilizer directly produced a negative effect on EF-GP, and indirectly reduced EF-GP by lowering soil pH. Nitrogen fertilizer also affected microbial diversity, which further negatively impacted EF-GP (*p* < 0.05). In conclusion, except for EF-N, nitrogen fertilizer addition directly exerted a negative effect on all ecosystem functions and multifunctionality in the structural equation models.

Random forest analysis was used to identify the effects of various explanatory variables, including soil and soil microbial factors, on ecosystem multifunctionality. The results of the random forest analysis model revealed that the most important explanatory factor for ecosystem multifunctionality was soil pH (Figure 8). This indicates that soil pH plays a crucial role in influencing the multifunctionality of the ecosystem.

### 3.4. Relationship Between Soil Microbial Communities and Ecological Functions

To explore the dominant ecological factors influencing the composition of soil bacterial and fungal communities, redundancy analysis (RDA) was performed, with the bacterial and fungal phyla as response variables and soil physicochemical factors as explanatory variables (Figure 9). RDA analysis of bacterial communities revealed that *Actinobacteriota* was positively correlated with EF-C, while *Gemmatimonadota*, *Chloroflexi*, *Firmicutes*, and *Methylomirabilota* were positively correlated with both EF-C and EF-N. In contrast, *Bacteroidota*, *Proteobacteria*, and *Armatimonadota* were positively correlated with EF-K and EF-P. RDA analysis of fungal communities showed that *Ascomycota* was positively correlated with EF-K and EF-P, while *Mortierellomycota*, *Basidiomycota*, and *Glomeromycota* were positively correlated with EF-C, EF-N, and EF-P.

## 4. Discussion

### 4.1. Nitrogen Fertilizer Addition Induced Soil Nutrients to Regulate Microbial Community

This study analyzed the microbial community structure in alfalfa soil under different nitrogen fertilizer additions and investigated the impact of nitrogen fertilizer on soil microorganisms. The results indicated a decline in the relative abundance of dominant microbial phyla with increasing nitrogen fertilizer application. This decline may be attributed to the increased soil acidity caused by excessive nitrogen fertilizer, which creates an unfavorable environment for certain microbial communities, thereby reducing their abundance [32]. Studies on soil microbial diversity have shown that bacterial diversity generally declines with increasing nitrogen fertilizer application, whereas fungal diversity tends to increase. This trend is primarily driven by the reduction in soil pH caused by nitrogen fertilization, as acidic conditions are typically unfavorable for bacterial growth, particularly for *Proteobacteria* and *Acidobacteriota*. In contrast, fungi, especially *Mortierellomycota*, thrive in acidic environments [33]. These findings align with the results of this study, where the relative abundance of *Proteobacteria* and *Acidobacteriota* was lowest under the N150 treatment, while the relative abundance of *Mortierellomycota* reached its highest under the same treatment. This further supports the notion that nitrogen-induced soil acidification selectively influences microbial community composition. PCA analysis revealed significant differences in bacterial community structure between the N50 treatment and other groups, indicating that the bacterial community was more sensitive to nitrogen fertilizer. There was no significant spatial variation in fungal communities between the N50, N150, and other treatments. This may be because bacteria occupy more active ecological niches in the soil and respond more quickly to nutrient changes, particularly nitrogen [34,35].

To clearly and intuitively illustrate the changes in microbial network relationships with nitrogen fertilizer addition and reveal its impact on microbial co-occurrence patterns, Gephi software was used to visualize the ecological networks for each treatment group. After data visualization, it was observed that bacterial communities in the CK treatment exhibited stronger competitive relationships among species, whereas bacterial communities in the N100 treatment showed stronger cooperative relationships. This could be attributed to bacterial species competing for limited resources (such as nutrients and space) in the absence of specific disturbances or regulatory factors. Such competition arises from resource non-renewability or factors like bacterial growth rates and reproductive strategies [36]. In the N100 treatment, nitrogen fertilizer addition may have promoted mutualistic or co-evolutionary relationships among species, enhancing overall community adaptability and stability, thereby improving their survival and reproduction under environmental changes or other stresses [37]. In contrast, the CK treatment exhibited a strong cooperative relationship among fungal species, likely due to complementary interactions in organic matter decomposition and nutrient cycling, which fostered cooperation within the community. Conversely, in the N50 treatment, fungal species exhibited a strong competitive relationship. This may be attributed to nitrogen addition creating specific constraints on fungal growth or reproduction, leading to resource limitations (e.g., nutrients and growth space), thereby intensifying competition for these scarce resources [38].

### 4.2. Nitrogen Fertilizer Addition Induces Soil Nutrient Regulation Ecosystem Multifunctionality

Increased nitrogen fertilizer addition affects plant growth, soil environments, and soil microorganisms, including plant growth parameters, soil physicochemical properties, and microbial community characteristics. These changes in turn influence the stability and multifunctionality of grassland ecosystems [39,40]. In this study, soil carbon nutrient, soil nitrogen nutrient, and ecosystem multifunctionality were higher in the N150 treatment compared to other treatments. This is likely because the N150 treatment provided a greater amount of nitrogen fertilizer, enhancing nitrogen absorption and utilization by plants, which in turn increased soil nitrogen nutrient levels and ecosystem multifunctionality [41]. Higher nitrogen fertilizer addition also promotes plant growth and biomass production, leading to greater carbon input into the soil and improved soil carbon nutrient levels [42]. Additionally, soil nutrient functions are determined by a combination of soil physical and chemical properties. The effect of nitrogen fertilizer addition on these properties ultimately translates into changes in ecosystem functions associated with the corresponding indicators.

This study investigated the mediating role of soil and soil microbial factors in ecosystem function and multifunctionality within grassland ecosystems under nitrogen fertilizer addition. Soil pH exhibited a significant negative effect on ecosystem multifunctionality, likely due to its influence on plant growth and species composition. Changes in soil pH can alter the structure and functionality of grassland ecosystems by favoring certain plant species while inhibiting others, ultimately leading to shifts in ecosystem multifunctionality. For example, acidic conditions may favor the growth of certain plant species while inhibiting others, leading to shifts in the multifunctionality of grassland ecosystems [43]. Soil pH has a significant negative impact on soil carbon nutrients and grassland productivity functions. This may be because soil acidity inhibits microbial growth and activity, reducing the decomposition of organic matter and consequently lowering soil carbon nutrient levels. Additionally, increased activity of aluminum and iron in acidic soils can be toxic to plant roots, further hindering plant growth and grassland productivity [44]. Soil water content has a significant negative effect on soil phosphorus nutrients, possibly due to excessive moisture causing soil hypoxia, which impairs root respiration and promotes the occurrence of diseases, ultimately reducing grassland productivity [45]. Microbial diversity has a significant positive effect on soil potassium nutrients but a significant negative effect on grassland productivity. This is likely because microbial communities efficiently fix and transform potassium in the soil, making it more readily available for plant absorption [46]. However, microbial activity may accelerate nutrient mineralization beyond the rate of plant uptake, thereby reducing grassland productivity [47].

### 4.3. Ecological Impacts and Limitations of This Study

This study investigated the effects of nitrogen fertilizer addition on soil ecosystem stability and microbial community composition in alfalfa fields using microbial co-occurrence networks and structural equation modeling. By examining soil ecosystem multifunctionality, we found that the N150 treatment had higher soil carbon and nitrogen content and greater ecosystem multifunctionality than other treatments, whereas soil potassium levels and grassland productivity were lower. Additionally, microbial interactions varied across treatments: bacterial communities in the N100 treatment exhibited strong cooperative relationships, while fungal communities in the N50 treatment showed intense competition. These findings underscore the need to assess whether different nitrogen fertilizer applications can effectively enhance soil ecosystem stability, thereby improving water retention and nutrient cycling efficiency [48]. Notably, research on soil microbial networks and ecosystem multifunctionality under varying nitrogen levels remains scarce. Our results indicate that increasing nitrogen fertilizer enhances microbial network connectivity and complexity, leading to greater ecosystem stability and resilience against environmental disturbances [49]. Moreover, higher soil ecosystem multifunctionality is associated with greater biodiversity, stronger nutrient cycling, and improved overall ecosystem stability [50]. These insights emphasize the crucial role of soil microorganisms and ecosystem multifunctionality in ecological sustainability and land management.

This study revealed the significant impact of nitrogen fertilizer addition on soil microorganisms and ecosystem multifunctionality in alfalfa fields on the Loess Plateau. The findings indicate that moderate nitrogen fertilizer addition can enhance microbial diversity and ecosystem multifunctionality. However, unreasonable nitrogen application can negatively affect microbial communities and ecosystem multifunctionality. Future research should focus on the integrated study of soil micro-food web structure and function in relation to soil ecosystem multifunctionality to gain a comprehensive understanding of the effects of nitrogen fertilizer addition on soil micro-food webs and ecosystem multifunctionality.

## 5. Conclusions

Our study presents strong evidence that nitrogen fertilizer addition directly exerted negative effects on ecosystem functions and multifunctionality. Soil pH significantly affected ecosystem multifunctionality, soil carbon nutrient functions, and grassland productivity functions, while soil water content significantly influenced soil phosphorus nutrient functions. Microbial diversity had significant impacts on grassland productivity functions and soil potassium nutrient functions. In summary, this study provides new insights into microbial community characteristics and ecosystem multifunctionality under different levels of nitrogen fertilizer addition in alfalfa fields. In the future, it will be important to monitor the long-term effects of nitrogen fertilizer addition on ecosystem functions and microbial communities.

## Figures and Tables

**Figure 1 microorganisms-13-00540-f001:**
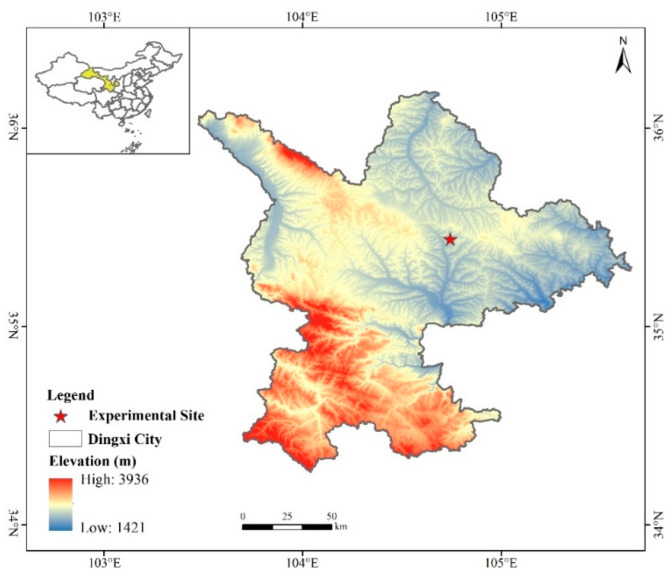
Location of the study area. (created by ArcGIS 10.2 software, http://desktop.arcgis.com/cn/, accessed on 1 February 2025).

**Figure 2 microorganisms-13-00540-f002:**
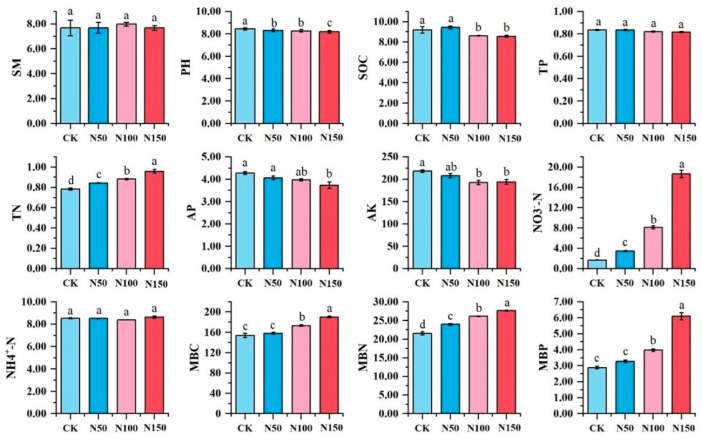
Physicochemical properties of soil under different nitrogen fertilizer treatments. Note: Different lowercase letters indicated that soil physical and chemical properties of different nitrogen treatments were significantly different (*p* < 0.05). SM-soil moisture; SOC-soil organic carbon; TN-total nitrogen; TP-total phosphorus; AP-available phosphorus; AK-available potassium; NO_3_-N-nitrate nitrogen; NH^4+^-N-ammonium nitrogen; MBC-microbial biomass carbon; MBN-microbial biomass nitrogen; MBP-microbial biomass phosphorus.

**Figure 3 microorganisms-13-00540-f003:**
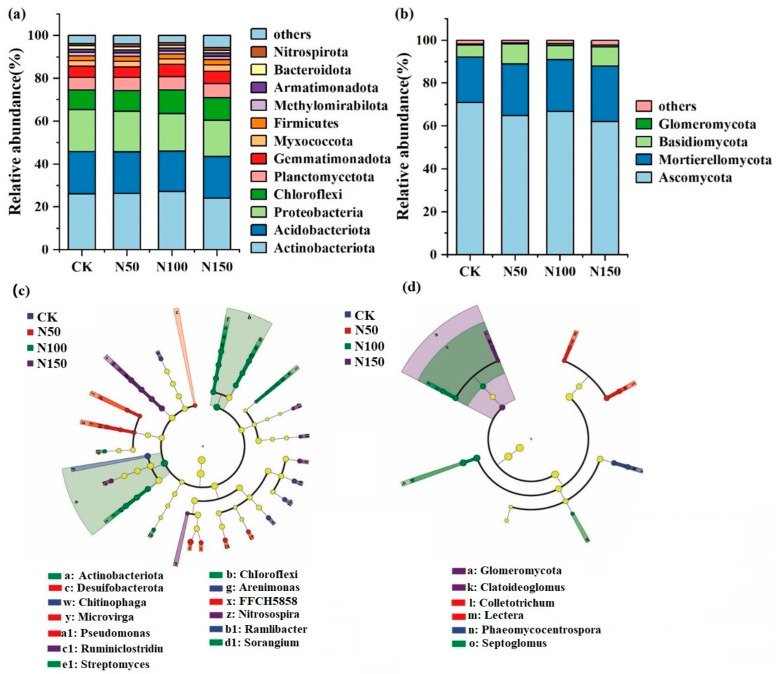
Composition of soil microbial communities under different nitrogen fertilizer treatments. (**a**): relative abundance of bacterial communities; (**b**): relative abundance of fungal communities; (**c**): Lefse analysis of the soil bacteria community; (**d**): Lefse analysis of the soil fungal community.

**Figure 4 microorganisms-13-00540-f004:**
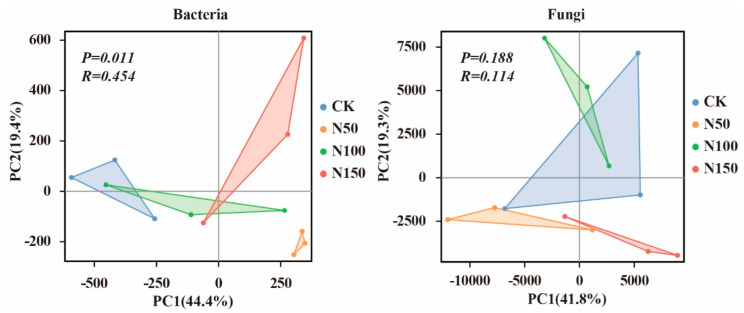
Principal Coordinate Analysis (PCA) of soil microbial communities under different nitrogen fertilizer treatments.

**Figure 5 microorganisms-13-00540-f005:**
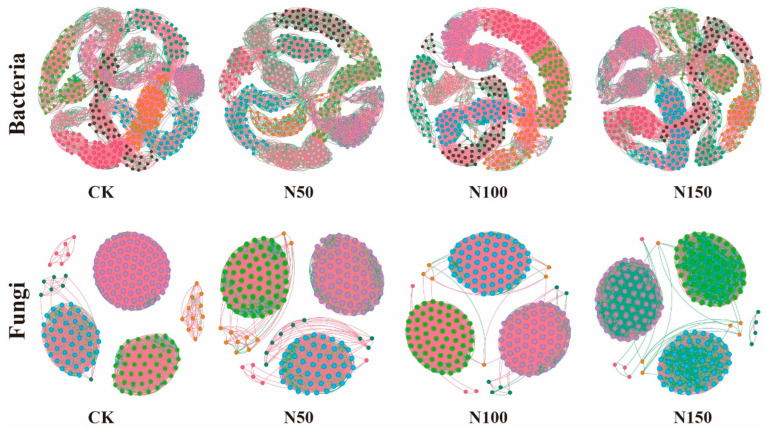
Co-occurrence network analysis of soil microbial communities under different nitrogen fertilizer treatments. Node size represents the degree (i.e., the number of connections to the node). Red edges indicate positive correlations, while green edges represent negative correlations. Nodes are colored according to their respective categories.

**Figure 6 microorganisms-13-00540-f006:**
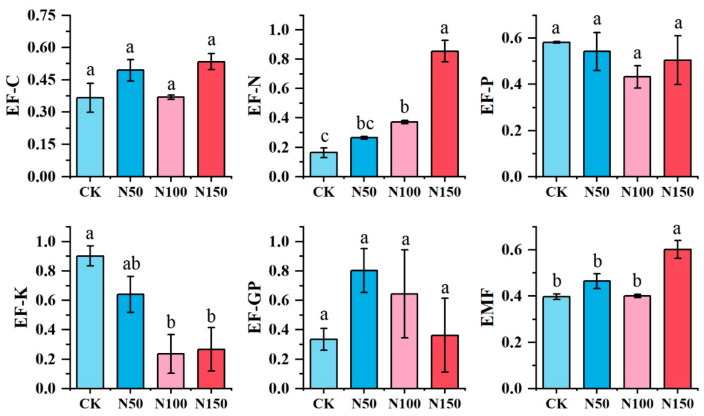
Multifunctionality of the soil ecosystem under different nitrogen fertilizer treatments. EF-GP: Alfalfa Primary Productivity; EF-C: Soil Carbon Nutrients and Storage; EF-N: Soil Nitrogen Nutrients and Storage; EF-P: Soil Phosphorus Nutrients and Storage; EF-K: Soil Potassium Nutrients and Storage; EMF: Soil Ecosystem Multifunctionality. Different lowercase letters indicate significant differences in multifunctionality of the soil ecosystem across the various nitrogen fertilizer treatments (*p* < 0.05). The same applies below.

**Figure 7 microorganisms-13-00540-f007:**
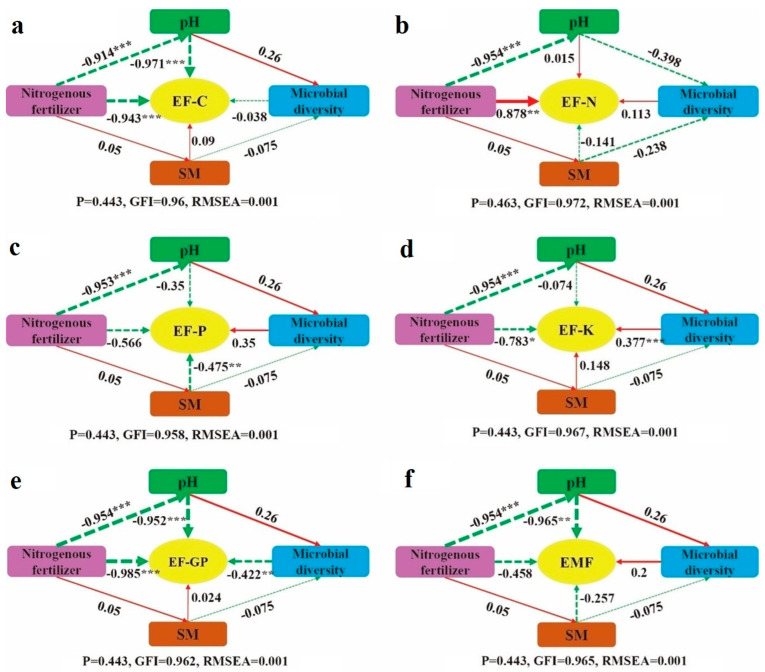
Structural equation model of nitrogen fertilizer addition on ecosystem function and multifunctionality. (**a**): EF-C structural equation; (**b**): EF-N structural equation; (**c**): EF-P structural equation; (**d**): EF-K structural equation; (**e**): EF-GP structural equation; (**f**): EMF structural equation. Note: Red solid arrows represent significant positive correlation paths, while green dashed arrows indicate negative correlation paths. The numbers on the arrows correspond to standardized path coefficients. Significance levels are indicated as * *p* < 0.05, ** *p* < 0.01, and *** *p* < 0.001. SM refers to soil water content.

**Figure 8 microorganisms-13-00540-f008:**
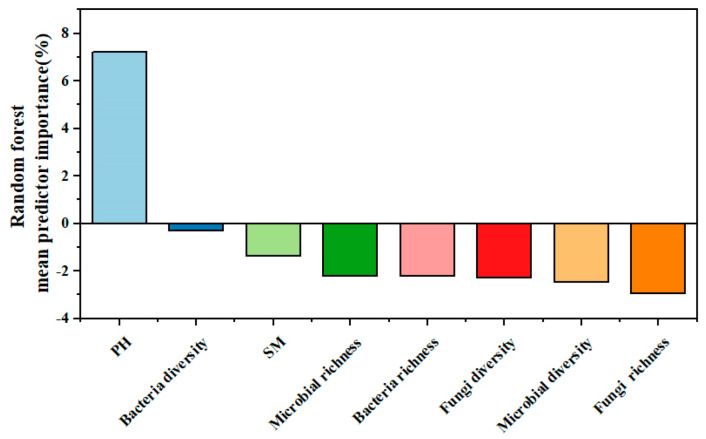
Random forest analysis.

**Figure 9 microorganisms-13-00540-f009:**
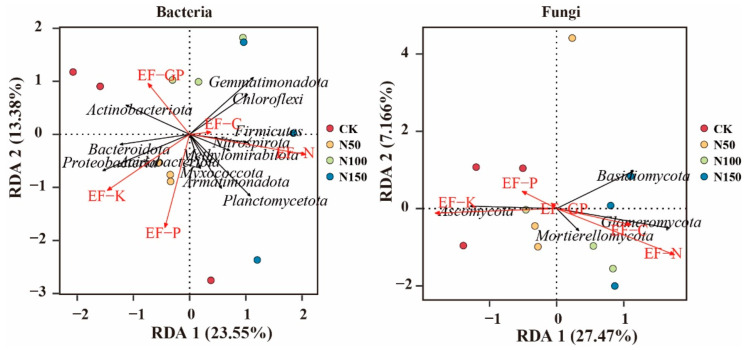
Relationship between bacterial and fungal phyla and soil ecosystem functions.

## Data Availability

The microbial DNA sequences from the 12 soil samples have been deposited in the Sequence Read Archive (SRA) of the NCBI database under Accession numbers NCBI: PRJNA1194357.

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
