# Peer review of "Nitrogen Fertilization Shapes Soil Microbial Diversity and Ecosystem Multifunctionality by Modulating Soil Nutrients"

_microorganisms, 2025, doi:10.3390/microorganisms13030540_

Round 1
Reviewer 1 Report
Comments and Suggestions for Authors
Dear Authors,
I have reviewed the manuscript and have the following observations.
The topic of the manuscript is soil microbial communities. The effects of nitrogen fertilizer application on ecosystem multifunctionality were investigated and it was found that changes in soil nutrient levels affect ecosystem multifunctionality by controlling microbial community diversity.
The topic of this manuscript is timely, as many agricultural and other soils have been destroyed by over-use of agriculture and fertilizers, resulting in a decline in soil life in many areas. Enriching this is an important task.
The manuscript is novel, as the thousandfold impact of nitrogen fertilization has been little discussed in journals.
The manuscript needs the following modifications:
The title is unnecessarily long, please draft and use a more pithy title
Abstract: it is too long and contains sections that should be in the Introduction chapter rather than here. Please shorten it to max 250 words is enough, in which Materials and methods, results and some figures should appear.
Discussion: reference 33 is not clickable and appears twice. The chapter is a little oversimplified - I suggest adding and comparing results of publications. Please apply the mind-boggling comparison as you move from the general to the specific.
Author Response
Dear Editor:
Thanks for your letter and for reviewer's comments concern our manuscript entitled “Nitrogen Fertilizer Addition Regulates Soil Microbial Community Diversity and Ecosystem Multifunctionality by Altering Soil Nutrient Dynamics in Alfalfa Fields on the Loess Plateau” (Manuscript ID: 3488218). Those comments are valuable and helpful for revising and improving our paper. We have studied all comments carefully and have made conscientious correction. Revised portion are marked in blue in the paper. The main corrections in the paper and the responds to the reviewer comments are as flowing.
Reviewer 1
I have reviewed the manuscript and have the following observations.
The topic of the manuscript is soil microbial communities. The effects of nitrogen fertilizer application on ecosystem multifunctionality were investigated and it was found that changes in soil nutrient levels affect ecosystem multifunctionality by controlling microbial community diversity. The topic of this manuscript is timely, as many agricultural and other soils have been destroyed by over-use of agriculture and fertilizers, resulting in a decline in soil life in many areas. Enriching this is an important task. The manuscript is novel, as the thousandfold impact of nitrogen fertilization has been little discussed in journals.
The manuscript needs the following modifications:
- The title is unnecessarily long, please draft and use a more pithy title.
Reply: Thank you for your valuable comments. We have revised the title accordingly; please refer to the updated manuscript for your review (Line 2-3).
2.Abstract: it is too long and contains sections that should be in the Introduction chapter rather than here. Please shorten it to max 250 words is enough, in which Materials and methods, results and some figures should appear.
Reply: Thank you for your valuable comments. We have revised the abstract accordingly; please refer to the updated manuscript for your review (Line 17-34).
- Discussion: reference 33 is not clickable and appears twice. The chapter is a little oversimplified - I suggest adding and comparing results of publications. Please apply the mind-boggling comparison as you move from the general to the specific.
Reply: Thank you for pointing this out. We have corrected the citation issue by ensuring that reference 33 is properly formatted and clickable. Additionally, we have removed the duplicate occurrence of this reference in the updated manuscript. please refer to the updated manuscript for your review (Line 399).
We acknowledge that the discussion was simplified. To strengthen this section, we have incorporated comparisons with recent and relevant publications. Specifically, we have compared our findings with studies that investigate the effects of nitrogen fertilization on soil microbial diversity and ecosystem multifunctionality. please refer to the updated manuscript for your review (Line 394-403, 422-428, 444-451,466-483).
Reviewer 2 Report
Comments and Suggestions for Authors
Microorganisms
Manuscript Draft
Manuscript Number: 34488218
Title: Nitrogen Fertilizer Addition Regulates Soil Microbial Community Diversity and Ecosystem Multifunctionality by Altering Soil Nutrient Dynamics in Alfalfa Fields on the Loess Plateau
Article Type: Research article
General Comments on MDPI Questions that Reviewers must answer:
- Is the manuscript clear, relevant for the field and presented in a well-structured manner?
This manuscript is written clearly, is very well-structured, and is potentially relevant to the field since it focuses on how increasing nitrogen fertilization impacts soil microbial communities in an alfalfa agricultural planting in the Loess Plateau in 2014. However, the manuscript needs to make the following SEVEN general improvements/clarifications:
1) Please clarify in the Methods section why only one year of the experiment is stated in 2014 when in the Introduction it is stated that this is an 8-year experiment. Which is it? Typical agronomic studies involve a minimum of two years to reduce variation from environmental and weather. An 8-year experiment is excellent. A one year experiment is not sufficient.
2) If there is only one year of data, please explain why the data being evaluated is from over 10 years ago. Why was this not published sooner? What sort of changes could be anticipated if the experiment was repeated today? If the experiment is an 8-year experiment, ignore this 2).
3) It is not clear from sub-section 2. in the 2. Materials and Methods section what was measured in what year. Was the soil sampled from 2014 through 2021? This needs to be clearly stated. Was soil microbial analyses conducted on samples from all eight years? Please also clarify in the writing for in the 3. Results section that the results are from measurements conducted over all eight years (if not, then please indicate otherwise).
4) The hypotheses in the last paragraph need to be moved to the Materials and Methods section. Replace this last paragraph with a paragraph that clearly states the goal(s) and objective(s) of the research.
5) Please add a new Figure 1 showing a map and/or pictures from the experiment.
6) Please add a Table 1 in the Results section of the parameters measured via soil sampling for each of the eight cropping years.
7) Please add a few sentences at the end of the paragraph in Conclusion indicating how future research can expand on the current study.
Given the potential contribution of this research, this manuscript has potential but requires more improvement to warrant publication in MDPI Microorganisms. Please also make the following minor edits and clarifications:
1) Please reduce the length of the title to increase clarity.
2) The Abstract is too long since it is greater than 200 words.
3) On L48-49, the keywords need to be in alphabetical order. The last keyword needs to follow the first and not be at the end.
4) The sub-headings need to have all major words capitalized (for the most part this is done but there are some exceptions such as on L245 where Diversity needs to be capitalized). The sub-headings need to be on one line so please shorten the ones that take up 2 lines such as on L389-390.
5) In sub-section 5. there are many words that do NOT need to be capitalized. Please correct this here and elsewhere.
6) All equations need a (#) all the way justified to the right and the equations need to be numbered in numerical order.
7) In the References, please add a period if a word in the journal name in italics is abbreviated so for example Soil Sci. Plant Nurtr. on L581 (use ISO4 abbreviations). The correct endash symbol (longer than a hyphen) needs to be used between the page ranges. Use the ; between co-authors and not a comma. There should be a space between the journal article name and the year. Remove the extra space between the year and the volume(issue). The volume(issue) needs to be in italics. There needs to be a comma after the volume(issue) and not a colon. Please add the DOI link at the end of each citation.
- Are the cited references mostly recent publications (within the last 5 years) and relevant? Does it include an excessive number of self-citations?
There are 24 of the 56 cited references have been published within the last 5 years and appear relevant to the research topic. There are no excessive self-citations.
- Is the manuscript scientifically sound and is the experimental design appropriate to test the hypothesis?
The manuscript is scientifically sound. The experimental analyses are appropriate.
- Are the manuscript’s results reproducible based on the details given in the methods section?
The manuscript’s experimental results are reproducible based on what is written in the 2. Materials and Methods section.
- Are the figures/tables/images/schemes appropriate? Do they properly show the data? Are they easy to interpret and understand? Is the data interpreted appropriately and consistently throughout the manuscript? Please include details regarding the statistical analysis or data acquired from specific databases.
The quality of the figures need slight improvement by more consistently applying a), b), c), d), etc. to individual graphs within each figure. Also, the a), b), c), d), etc. needs to be added to each figure caption if there are multiple graphs.
- Are the conclusions consistent with the evidence and arguments presented?
The Conclusions are consistent with the evidence and arguments presented. Please add a few sentences at the end of the Conclusions section on how future research can improve on the current work.
- Please evaluate the data availability statements to ensure it is adequate.
All Back Matter sections are fine with the exception that the Acknowledgements section need to be added in the proper location.
Author Response
Dear Editor:
Thanks for your letter and for reviewer's comments concern our manuscript entitled “Nitrogen Fertilizer Addition Regulates Soil Microbial Community Diversity and Ecosystem Multifunctionality by Altering Soil Nutrient Dynamics in Alfalfa Fields on the Loess Plateau” (Manuscript ID: 3488218). Those comments are valuable and helpful for revising and improving our paper. We have studied all comments carefully and have made conscientious correction. Revised portion are marked in blue in the paper. The main corrections in the paper and the responds to the reviewer comments are as flowing.
Reviewer 2
This manuscript is written clearly, is very well-structured, and is potentially relevant to the field since it focuses on how increasing nitrogen fertilization impacts soil microbial communities in an alfalfa agricultural planting in the Loess Plateau in 2014. However, the manuscript needs to make the following SEVEN general improvements/clarifications:
1) Please clarify in the Methods section why only one year of the experiment is stated in 2014 when in the Introduction it is stated that this is an 8-year experiment. Which is it? Typical agronomic studies involve a minimum of two years to reduce variation from environmental and weather. An 8-year experiment is excellent. A one year experiment is not sufficient.
Reply: Thank you for your comment. We established the experiment in 2014, collecting initial soil samples before the study began to assess soil physical and chemical properties as well as aboveground biomass indicators. However, as a perennial forage, alfalfa exhibited minimal yield differences in the first few years after sowing, making the early effects of fertilization less apparent. After eight years of continuous experimentation, the mid-term fertilization effects became more pronounced. Therefore, in 2021, we collected soil samples to analyze microbial communities, allowing us to explore the long-term impacts of alfalfa cultivation and fertilization on soil health. To further assess long-term fertilization effects, we plan to conduct another round of sampling in 2026. We appreciate your valuable review and feedback and look forward to further discussions. We hope our research contributes meaningful insights into soil ecosystem health and stability.
2) If there is only one year of data, please explain why the data being evaluated is from over 10 years ago. Why was this not published sooner? What sort of changes could be anticipated if the experiment was repeated today? If the experiment is an 8-year experiment, ignore this 2).
Reply: Thank you for your insightful comments. I haven't published it for personal reasons. Alfalfa is a perennial nitrogen fixing plant, and the yield difference in previous years was not significant. Therefore, we took the yield, physical and chemical properties and microorganisms of alfalfa in 8 years as the dividing points. In this phase, our research explores more deeply the long-term effects of alfalfa planting and fertilization on soil health. In the future, we plan to conduct further sampling in 2026 to assess long-term fertilization effects.
3) It is not clear from sub-section 2. in the 2. Materials and Methods section what was measured in what year. Was the soil sampled from 2014 through 2021? This needs to be clearly stated. Was soil microbial analyses conducted on samples from all eight years? Please also clarify in the writing for in the 3. Results section that the results are from measurements conducted over all eight years (if not, then please indicate otherwise).
Reply: Thank you for your valuable feedback. We apologize for any confusion. To clarify, initial soil sampling was conducted in 2014 before the experiment began, primarily to assess baseline aboveground biomass indicators. However, soil microbial analyses were not performed annually. Instead, we conducted microbial analysis on soil samples collected in 2021, as this was the point at which the mid-term effects of fertilization had fully manifested. To ensure clarity, we will revise Section 2 (Materials and Methods) to explicitly state which indicators were measured in each year. Additionally, in Section 3 (Results), we will clearly specify that the results are based on the 2021 sampling rather than continuous measurements over all eight years. Please refer to the updated manuscript for your review (Line 140, 147-148). We appreciate your constructive suggestions and will refine the manuscript accordingly to enhance transparency and readability.
4) The hypotheses in the last paragraph need to be moved to the Materials and Methods section. Replace this last paragraph with a paragraph that clearly states the goal(s) and objective(s) of the research.
Reply: Thank you for your valuable suggestion. We hypotheses are generally in the introduction, so we have not moved them to the materials and methods section. However, where hypotheses are made, we rearrange them to clearly state the goal and purpose of the research. Specifically, this study aims to explore the impact of nitrogen fertilizer addition on soil ecosystem stability and microbial community composition in alfalfa fields. By analyzing soil physicochemical properties, microbial communities, and ecosystem multifunctionality, we seek to understand the mechanisms through which nitrogen fertilization influences soil health and ecosystem functions. The revised manuscript reflects these changes (see Line 100-116). We appreciate your insightful feedback and look forward to any further suggestions to improve the clarity and impact of our research.
5) Please add a new Figure 1 showing a map and/or pictures from the experiment.
Reply: Thank you very much for your review and detailed feedback on our manuscript. In response to your suggestion, we have added a new Figure 1 showing a map from the experiment. Please refer to the updated manuscript for your review (Figure 1).
6) Please add a Table 1 in the Results section of the parameters measured via soil sampling for each of the eight cropping years.
Reply: Thank you very much for your valuable comments. Since the soil measurement parameter analysis in this study was only conducted in 2021, we are unable to provide specific data for each year. Therefore, Table 1 will only contain soil measurement parameter data for 2021. Additionally, we have added a new Figure 2 in the Results section of the paper, which shows the measurement parameters for each soil sampling point in 2021. Please refer to the revised manuscript for further details.
7) Please add a few sentences at the end of the paragraph in Conclusion indicating how future research can expand on the current study.
Reply: Thank you very much for your valuable comments. We have made revisions as requested. Please see the revised manuscript for details (Line 501-503).
Given the potential contribution of this research, this manuscript has potential but requires more improvement to warrant publication in MDPI Microorganisms. Please also make the following minor edits and clarifications:
1) Please reduce the length of the title to increase clarity.
Reply: Thank you for your valuable comments. We have revised the title accordingly; please refer to the updated manuscript for your review (Line 2-3).
2) The Abstract is too long since it is greater than 200 words.
Reply: Thank you for your valuable comments. We have revised the abstract accordingly; please refer to the updated manuscript for your review (Line 17-34).
3) On L48-49, the keywords need to be in alphabetical order. The last keyword needs to follow the first and not be at the end.
Reply: Thank you for your valuable comments. We have revised the abstract keywords accordingly; please refer to the updated manuscript for your review (Line 35-36).
4) The sub-headings need to have all major words capitalized (for the most part this is done but there are some exceptions such as on L245 where Diversity needs to be capitalized). The sub-headings need to be on one line so please shorten the ones that take up 2 lines such as on L389-390.
Reply: Thank you for your valuable comments. We have revised all sub-headings accordingly; please refer to the updated manuscript for your review (Line 244, 322, 387, 429, 465).
5) In sub-section 5. there are many words that do NOT need to be capitalized. Please correct this here and elsewhere.
Reply: Thank you for pointing this out. We have reviewed sub-section 5 and other sections of the manuscript, and we have made the necessary corrections to ensure that only the appropriate words are capitalized. please refer to the updated manuscript for your review (Line189-198).
6) All equations need a (#) all the way justified to the right and the equations need to be numbered in numerical order.
Reply: Thank you for this suggestion. We have carefully reviewed all equations in the manuscript and have made the following changes: Each equation has been numbered consecutively in numerical order, as per the formatting guidelines. please refer to the updated manuscript for your review (Line204, 212, 214).
7) In the References, please add a period if a word in the journal name in italics is abbreviated so for example Soil Sci. Plant Nurtr. on L581 (use ISO4 abbreviations). The correct endash symbol (longer than a hyphen) needs to be used between the page ranges. Use the; between co-authors and not a comma. There should be a space between the journal article name and the year. Remove the extra space between the year and the volume(issue). The volume(issue) needs to be in italics. There needs to be a comma after the volume(issue) and not a colon. Please add the DOI link at the end of each citation.
Reply: Thank you for your detailed feedback on the References section. We have made the following revisions: added a period after abbreviated journal names, replaced hyphens with the correct endash symbol between page ranges, changed commas to semicolons between co-authors, ensured proper spacing between the article title and the year, removed extra space between the year and volume/issue, italicized the volume(issue), and added a comma after it instead of a colon. Additionally, we have included DOI links at the end of each citation. These revisions ensure our references conform to the journal's citation style.
Round 2
Reviewer 1 Report
Comments and Suggestions for Authors
I recommand it for publication.
Author Response
Dear Editor:
Thanks for your letter and for reviewer's comments concern our manuscript entitled “Nitrogen Fertilizer Addition Regulates Soil Microbial Community Diversity and Ecosystem Multifunctionality by Altering Soil Nutrient Dynamics in Alfalfa Fields on the Loess Plateau” (Manuscript ID: 3488218). Those comments are valuable and helpful for revising and improving our paper. We have studied all comments carefully and have made conscientious correction. Revised portion are marked in blue in the paper. The main corrections in the paper and the responds to the reviewer comments are as flowing.
Reviewer 1
I recommand it for publication.
Reply:Thank you very much for your positive feedback and for recommending our manuscript for publication. We greatly appreciate your time and effort in reviewing our work, and we are pleased to hear that you found it suitable for publication.
We have carefully considered all the comments and suggestions provided by you and the other reviewers, and we have made the necessary revisions to improve the manuscript. We believe these changes have strengthened the paper, and we hope that the revised version meets your expectations.
Once again, thank you for your valuable input and support.
Reviewer 2 Report
Comments and Suggestions for Authors
Microorganisms
Manuscript Draft
Manuscript Number: 34488218
Title: Nitrogen Fertilization Shapes Soil Microbial Diversity and Ecosystem Multifunctionality by Modulating Soil Nutrients
Article Type: Research article
Please make the following final minor edits:
1) On first page, summary in left margin is missing (add back from Word template).
2) Do not use abbreviations in Abstract so write out what CK stands for.
3) With the exception of the first keyword Alfalfa, all other keywords should NOT be capitalized.
4) L130-131 should be left justified and not center justified.
5) On L133, write as 6 April 2014.
6) On L150-163, there is no need to use abbreviations if they are not used afterwards. Please make sure this is followed throughout the manuscript.
7) Do not indent L213.
8) For major sections, the larger font size for example on L386 is correct. Please make sure this is consistently done throughout the manuscript (e.g., Results).
9) On L429 and L465, capitalize all major words so this is consistent.
10) In the References, please use ISO4 abbreviations for ALL journal articles cited (this can be done through a simple online search. This was not done for all. For example on L532, this should be Tot. Environ.
Author Response
Dear Editor:
Thanks for your letter and for reviewer's comments concern our manuscript entitled “Nitrogen Fertilizer Addition Regulates Soil Microbial Community Diversity and Ecosystem Multifunctionality by Altering Soil Nutrient Dynamics in Alfalfa Fields on the Loess Plateau” (Manuscript ID: 3488218). Those comments are valuable and helpful for revising and improving our paper. We have studied all comments carefully and have made conscientious correction. Revised portion are marked in blue in the paper. The main corrections in the paper and the responds to the reviewer comments are as flowing.
Reviewer 2
Please make the following final minor edits:
1) On first page, summary in left margin is missing (add back from Word template).
Reply: Thank you for your valuable comments. We have added a summary in the left margin, please refer to the updated manuscript for your review (page 1).
2) Do not use abbreviations in Abstract so write out what CK stands for.
Reply: Thank you for your valuable comments. We have revised the CK accordingly, please refer to the updated manuscript for your review (Line 24-25).
3) With the exception of the first keyword Alfalfa, all other keywords should NOT be capitalized.
Reply: Thank you for your valuable comments. We have revised the abstract keywords accordingly; please refer to the updated manuscript for your review (Line 35-36).
4) L130-131 should be left justified and not center justified.
Reply: Thank you for your valuable comments. We have revised it, please refer to the updated manuscript for your review (Line 130-131).
5) On L133, write as 6 April 2014.
Reply: Thank you for your valuable comments. We have revised it, please refer to the updated manuscript for your review (Line 133).
6) On L150-163, there is no need to use abbreviations if they are not used afterwards. Please make sure this is followed throughout the manuscript.
Reply: Thank you for your valuable comments. We have revised the abbreviations accordingly, please refer to the updated manuscript for your review (Line 150-164).
7) Do not indent L213.
Reply: Thank you for your valuable comments. We have revised it, please refer to the updated manuscript for your review (Line 214).
8) For major sections, the larger font size for example on L386 is correct. Please make sure this is consistently done throughout the manuscript (e.g., Results).
Reply: Thank you for your valuable comments. We have revised the results accordingly, please refer to the updated manuscript for your review (Line 230).
9) On L429 and L465, capitalize all major words so this is consistent.
Reply: Thank you for your valuable comments. We have revised the major words accordingly,please refer to the updated manuscript for your review (Line 433 and Line 469).
10) In the References, please use ISO4 abbreviations for ALL journal articles cited (this can be done through a simple online search. This was not done for all. For example on L532, this should be Tot. Environ.
Reply: Thank you for your valuable comments. We have revised the references accordingly, please refer to the updated manuscript for your review.